# Impact of chronic oral glucocorticoid treatment on mortality in patients with COVID-19: analysis of a population-based cohort

Margret J Einarsdottir [1,2] Brian Kibiwott Kirui [3] Huiqi Li [3] Daniel Olsson [1,2,4] Gudmundur Johannsson [1,2] Fredrik Nyberg [3] Oskar Ragnarsson [1,2,5]

MJE and BKK are joint first authors.

FN and OR are joint senior authors.

For numbered affiliations see end of article.

**Correspondence to**
Dr Oskar Ragnarsson;
oskar.ragnarsson@medic.gu.se

## ABSTRACT

**Objectives** While glucocorticoid (GC) treatment initiated for COVID-19 reduces mortality, it is unclear whether GC treatment prior to COVID-19 affects mortality. Long-term GC use raises infection and thromboembolic risks. We investigated if patients with oral GC use prior to COVID-19 had increased mortality overall and by selected causes.

**Design** Population-based observational cohort study.

**Settings** Population-based register data in Sweden.

**Participants** All patients infected with COVID-19 in Sweden from January 2020 to November 2021 (n=1 200 153).

**Outcome measures** Any prior oral GC use was defined as ≥1 GC prescription during 12 months before index. High exposure was defined as ≥2 GC prescriptions with a cumulative prednisolone dose ≥750 mg or equivalent during 6 months before index. GC users were compared with COVID-19 patients who had not received GCs within 12 months before index. We used Cox proportional hazard models and 1:2 propensity score matching to estimate HRs and 95% CIs, controlling for the same confounders in all analyses.

**Results** 3378 deaths occurred in subjects with any prior GC exposure (n=48 806; 6.9%) and 14 850 among non-exposed (n=1 151 347; 1.3%). Both high (HR 1.98, 95% CI 1.87 to 2.09) and any exposure (1.58, 1.52 to 1.65) to GCs were associated with overall death. Deaths from pulmonary embolism, sepsis and COVID-19 were associated with high GC exposure and, similarly but weaker, with any exposure. High exposure to GCs was associated with increased deaths caused by stroke and myocardial infarction.

**Conclusion** Patients on oral GC treatment prior to COVID-19 have increased mortality, particularly from pulmonary embolism, sepsis and COVID-19.

## STRENGTHS AND LIMITATIONS OF THIS STUDY

⇒ Uses a population-based design.
⇒ Employs adjusted Cox models and propensity score matching for confounder control.
⇒ A large number of covariates, and a sizeable comparison cohort were used.
⇒ Observational nature may still introduce residual bias and unmeasured confounding.

## INTRODUCTION

Oral glucocorticoids (GCs) are commonly prescribed for a variety of conditions, including inflammatory disorders as well as various allergic, respiratory, skin and autoimmune conditions.[1] In Sweden, 0.5% of the population receive long-term (>2 years) oral GC treatment,[1] a treatment that is known to be associated with excess mortality mainly due to myocardial infarction, sepsis and pulmonary embolism.[2] Hypercortisolism is associated with a hypercoagulable state due to increased levels of procoagulant factors and impaired fibrinolysis.[3] Patients receiving long-term GC treatment frequently also develop GC-induced adrenal insufficiency that renders them at increased risk of adrenal crisis, and subsequent death, during a severe infection.[4]

Thromboembolic events, including pulmonary embolism and stroke, are common complications in patients with COVID-19.[5] Despite anticoagulation treatment, a notable proportion (16.7%) of patients with COVID-19 develop pulmonary embolism.[5] COVID-19 patients may also develop arterial thrombosis, which can lead to serious outcomes such as stroke or myocardial infarction.[6]

The use of oral GCs in COVID-19 patients has been a topic of research and debate. Current literature on GC therapy initiated for COVID-19 and effect on thromboembolic events warns that GC administration during COVID-19 may increase the risk of thromboembolic events, particularly in critically ill patients or those with pre-existing thrombotic risk factors.[7] In May 2020, WHO published

clinical guidelines on the management of COVID-19 and advised against the use of GCs unless indicated for another reason (eg, exacerbation of asthma or chronic obstructive pulmonary disease).[8] At that time, high-dose GC therapy initiated for COVID-19 had been associated with delayed virus clearing, prolonged hospitalisation and increased use of mechanical ventilation in patients with COVID-19.[9] In July 2020, the Randomized Evaluation of COVID-19 Therapy (RECOVERY) trial showed that dexamethasone treatment initiated in COVID-19 patients resulted in lower 28-day mortality among those requiring respiratory support (mechanical ventilation or oxygen alone).[10] Consequently, in September 2020, WHO revised the guidelines and recommended initiation of systemic GC treatment for patients with severe and critical COVID-19.[11] Nevertheless, GC treatment for COVID-19 patients is still controversial. GC therapy initiated for COVID-19 was related to increased death compared with conventional treatment in many observational studies, while beneficial effects on mortality were seen in randomised clinical trials.[12] There is also a concern regarding the dose of GC therapy initiated for COVID-19, as higher doses have been shown to increase mortality rates in hypoxic patients not requiring ventilatory support.[13] In the aforementioned studies, GC treatment was initiated for the treatment of COVID-19.

Not much is known about how long-term GC treatment prior to COVID-19 effects outcome and mortality. An international study of patients with rheumatic disease receiving long-term GC treatment showed that prednisolone ≥10 mg/day prior to COVID-19 was associated with an increased hospitalisation rate.[14] Chronic systemic GC therapy has also been associated with increased mortality in immunosuppressed patients with COVID-19.[15] Whether patients receiving GC treatment for other conditions are at increased risk of death after developing COVID-19 has not been systematically studied.

Our hypothesis was that patients on oral GC treatment before being diagnosed with COVID-19 have an increased mortality from several conditions, including pulmonary embolism, myocardial infarction, stroke, sepsis and COVID-19 itself compared with those who do not have GC treatment before COVID-19. To test this hypothesis, we investigated the association between prior GC treatment and the risk of mortality from these conditions among COVID-19 patients using population-based analysis.

## METHODS

### Study design, data sources and study population

This was a register-based cohort study centred on the regularly updated Swedish COVID-19 Investigation for Future Insights—a Population Epidemiology Approach using Register Linkage (SCIFI-PEARL) project database,[16] which currently includes the entire Swedish population from 2015.

The study population consisted of individuals with COVID-19 who met the following criteria before 30 November 2021: (a) a positive test for SARS-CoV-2 in the National Database of Notifiable Communicable Diseases (SmiNet); (b) identified in the Swedish National Patient Register (NPR) with the International Classification of Diseases, revision 10 (ICD-10) codes U07.1 or U07.2 from a specialist visit or hospitalisation or (c) identified in the Cause-of-Death Register with COVID-19 as an underlying or contributing cause of death. For each individual, their earliest date of meeting any of these COVID-19 diagnosis criteria was considered their index date (baseline) in the study.

Additional data and sources used from the SCIFI-PEARL database included: sociodemographic data and dates of emigration or death from Statistics Sweden (Register of Total Population and the Longitudinal Integrated Database for Health Insurance and Labor Market Studies)[17]; prescribed medication data from the National Prescribed Drug Register (NPDR)[18]; comorbidities from the NPR[19] and intensive care data from the Swedish Intensive Care Register (SIR). Comprehensive data for each patient were obtained by linking their information across the different registers using the national personal identification number, as previously described.[16]

### Patient and public involvement

Patients and/or the public were not involved in the design, or conduct, or reporting, or dissemination plans of this research.

### Exposures, outcomes and follow-up

Data from the NPDR were used to identify oral GC users as well as GC dosage and formulation (capsules, drops or tablets). Recent or ongoing oral GC treatment was defined in two ways: (a) any prior exposure as ≥1 dispensed prescriptions of oral GCs within 12 months before the index date and (b) high exposure defined >2 dispensed prescriptions with a cumulative dose of prednisolone (Anatomical Therapeutic Chemical code H02AB06) ≥750 mg, or equivalent amount (online supplemental table S1) of betamethasone (H02AB01), dexamethasone (H02AB02), methylprednisolone (H02AB04), prednisone (H02AB07), hydrocortisone (H02AB09), cortisone (H02AB10), deflazacort (H02AB13) or a combination of these within 6 months before the index date. All patients belonging to the 'high exposure' group were thus also included in the 'any prior exposure' group. These exposure definitions target recent use of oral GC, and for the 'high exposure' more consistent and recent use, more likely extending into the future follow-up period, thus sharpening the identification of regular users and as such, patients at risk, and ensuring robust data analysis. The non-exposed comparison group included COVID-19 patients who had not received any oral GC prescriptions during the year before their index date. High exposure was defined as prednisolone ≥750 mg within 6 months before the index date, that is, approximately ≥4–5 mg of prednisolone per day which is supraphysiological dose of

GC.[20] Doses <5 mg of prednisolone per day have not been associated with an increased risk of death.[21]

The primary outcomes were overall death (all-cause mortality) and cause-specific death according to ICD-10 classification due to pulmonary embolism (I26), sepsis (A40, A41 or R57.2), myocardial infarction (I21–I23), stroke (I60–I64, I630–I635, I638 or I639) and COVID-19 (U07.1 or U07.2). Outcomes of cause-specific death were identified using the Cause-of-Death Register with the diagnosis being either the underlying or contributing cause. Secondary outcomes were two measures of severe COVID-19, hospitalisation for COVID-19 and direct admission to an intensive care unit (ICU), each defined by U07.1 or U07.2 as primary or secondary diagnosis, respectively, in the NPR or SIR. The different outcomes were all analysed separately. For each outcome, all subjects were followed from the index date (COVID-19 infection date) until the earliest of the following: an event for the specific outcome, emigration, death or end of the study period (30 November 2021).

### Baseline covariates

The following baseline information was extracted from relevant registers (for details see online supplemental table S2): (1) demographic factors from Statistics Sweden including age, sex, education and employment; (2) comorbidities from the NPR as primary or secondary diagnoses for specialist outpatient or inpatient care from 1 January 2015 until the index date and (3) dispensed prescribed medications from the NPDR during the year before the index date. Based on a priori selection, the following covariates were used for confounding adjustment: age, sex, education, employment and relevant previous medical history (diabetes, deep vein thrombosis, pulmonary embolism, hypertension, stroke, ischaemic heart disease, heart failure, cancer, chronic obstructive pulmonary disease, rheumatological conditions and oral anticoagulation therapy). In sensitivity analyses, we considered additional covariates: adrenal insufficiency, rheumatoid arthritis, chronic lower respiratory diseases, other respiratory diseases principally affecting the interstitium, non-infective enteritis and colitis, dermatitis and eczema, inflammatory polyarthropathies, systemic connective tissue disorders (except rheumatoid arthritis), glomerular diseases, organ or tissue transplant, chronic liver disease, alcoholism and other substance abuse and the following medications: inhaled corticosteroids, statins, ACE inhibitors (ACEIs) and opioids (online supplemental table S2).

### Statistical analysis

The exposed and non-exposed groups were characterised using descriptive statistics with continuous variables presented as mean and SD, and categorical variables as number (%). Differences between exposed and non-exposed groups were characterised using standardised mean differences (SMDs). Kaplan-Meier curves were used to examine the outcomes over time during the study period. Cox proportional hazards regression was used to obtain hazard ratios (HRs) with 95% confidence intervals (CIs). In the main analysis, confounding was addressed by adjusting for the selected potential confounders in standard Cox models. In an alternative analysis approach, confounding was addressed by 1:2 matching on a propensity score that included the same set of confounders as in the main regression analysis. The propensity score was estimated by fitting a logistic regression model with the selected covariates as predictors and oral GC exposure as independent variables. A 1:2 matching on the propensity score was then performed and evaluated by assessment of SMD prematching and postmatching for all variables and by visually checking the propensity score overlap. Sensitivity analyses were also conducted incorporating additional potential confounders of interest, in particular greater detail regarding the underlying indications for oral GC. This sensitivity analysis could, however, instead suffer from potential overadjustment in relation to the study hypothesis of understanding the overall increased risk patients in these groups on oral GC faced when infected with COVID-19. Analyses were performed in SAS version 9.4, and R statistical software V.4.04 and matching was implemented using the MatchIT package in R.[22]

## RESULTS

### Demographic and clinical characteristics

A cohort of 1 200 153 individuals with COVID-19 was identified with 48 806 having any GC exposure the year before index, including 13 493 individuals with high GC exposure during the last 6 months (table 1). The exposed groups were older and included more women, whereas the non-exposed group had a more even sex distribution. Substantial differences between the exposed and non-exposed (SMDs >0.2) were found for age, sex and multiple comorbidities including cardiovascular, respiratory, cancer and rheumatic disease (table 1). There were 3378 deaths in patients with any GC exposure (6.9%), of which 2023 deaths occurred in patients with high exposure (15.0%) and 14 850 deaths in the non-exposed group (1.3%) (online supplemental table S3).

### Association between GC exposure and death

Unadjusted cumulative incidence curves indicated that crude overall and cause-specific mortality was higher among GC users compared with non-users (figures 1 and 2).

In the main Cox regression analyses, the adjusted HR for all-cause mortality was 1.98 (95% CI 1.87 to 2.09) in the high exposure group and 1.58 (1.52 to 1.65) in patients with any exposure compared with the non-exposed group (figure 3). High exposure to oral GCs was also associated with significantly increased risks of death from pulmonary embolism, sepsis, stroke and COVID-19. Any prior exposure was similarly associated with death due to pulmonary embolism, sepsis and COVID-19, but not significantly with fatal stroke (figure 3). The

**Table 1** Baseline characteristics of individuals with COVID-19 infection in Sweden from January 2020 to November 2021 without and with glucocorticoid exposure prior to the infection

| | Non-exposed (n=1 151 347) | Any prior GC exposure* (n=48 806) | | High GC exposure† (n=13 497) | |
|---|---|---|---|---|---|
| | | | SMD versus non-exposed | | SMD versus non-exposed |
| Age (years) | 37.9 (19.2) | 51.4 (21.2) | 0.666 | 62.4 (19.7) | 1.257 |
| Age category (years) | | | 0.672 | | 1.276 |
| 0–9 | 56 461 (4.9%) | 930 (1.9%) | | 91 (0.7%) | |
| 10–19 | 171 590 (14.9%) | 3056 (6.3%) | | 284 (2.1%) | |
| 20–29 | 200 477 (17.4%) | 4491 (9.2%) | | 567 (4.2%) | |
| 30–39 | 200 414 (17.4%) | 5882 (12.1%) | | 925 (6.9%) | |
| 40–49 | 205 427 (17.8%) | 8045 (16.5%) | | 1514 (11.2%) | |
| 50–59 | 167 108 (14.5%) | 8993 (18.4%) | | 2249 (16.7%) | |
| 60–69 | 79 704 (6.9%) | 6396 (13.1%) | | 188 (16.2%) | |
| 70–79 | 37 797 (3.3%) | 5725 (11.7%) | | 2680 (19.9%) | |
| 80–89 | 23 055 (2.0%) | 4123 (8.4%) | | 2302 (17.1%) | |
| ≥90 | 9314 (0.8%) | 1165 (2.4%) | | 697 (5.2%) | |
| Sex | | | 0.149 | | 0.147 |
| Male | 566 579 (49.2%) | 20 407 (41.8%) | | 5659 (41.9%) | |
| Female | 584 768 (50.8%) | 28 399 (58.2%) | | 7838 (58.1%) | |
| Birth country | | | 0.105 | | 0.199 |
| Sweden | 883 118 (76.7%) | 38 515 (78.9%) | | 10 923 (80.9%) | |
| Nordic | 18 577 (1.6%) | 1285 (2.6%) | | 484 (3.6%) | |
| Europe except Nordics | 83 455 (7.2%) | 3142 (6.4%) | | 779 (5.8%) | |
| Other | 166 197 (14.4%) | 5864 (12.0%) | | 1311 (9.7%) | |
| Employment status | | | 0.453 | | 0.888 |
| Employed | 838 341 (72.8%) | 31 232 (64.0%) | | 6432 (47.7%) | |
| Unemployed | 172 951 (15.0%) | 15 328 (31.4%) | | 6857 (50.8%) | |
| Unknown | 140 055 (12.2%) | 2246 (4.6%) | | 208 (1.5%) | |
| Education level | | | 0.300 | | 0.501 |
| Primary | 173 194 (15.0%) | 9817 (20.1%) | | 3672 (27.2%) | |
| Secondary | 426 657 (37.1%) | 19 950 (40.9%) | | 5656 (41.9%) | |
| Higher | 376 185 (32.7%) | 15 855 (32.5%) | | 3714 (27.5%) | |
| Unknown | 175 311 (15.2%) | 3184 (6.5%) | | 455 (3.4%) | |
| Comorbidities | | | | | |
| Pulmonary embolism | 7977 (0.7%) | 1676 (3.4%) | 0.194 | 853 (6.3%) | 0.310 |
| Stroke | 12 140 (1.1%) | 1679 (3.4%) | 0.161 | 852 (6.3%) | 0.282 |
| Deep vein thrombosis | 1085 (0.1%) | 301 (0.6%) | 0.088 | 97 (0.7%) | 0.098 |
| Hypertension | 86 847 (7.5%) | 13 486 (27.6%) | 0.547 | 6255 (46.3%) | 0.972 |
| Ischaemic heart disease | 26 541 (2.3%) | 4370 (9.0%) | 0.291 | 2147 (15.9%) | 0.487 |
| Overall CV disease | 149 594 (13.0%) | 19 033 (39.0%) | 0.621 | 8293 (61.4%) | 1.158 |
| Heart failure | 18 043 (1.6%) | 4456 (9.1%) | 0.341 | 2347 (17.4%) | 0.561 |
| Asthma | 37 185 (3.2%) | 7841 (16.1%) | 0.445 | 2026 (15.0%) | 0.418 |
| COPD | 8731 (0.8%) | 3856 (7.9%) | 0.356 | 1700 (12.6%) | 0.488 |
| Emphysema | 793 (0.1%) | 392 (0.8%) | 0.112 | 199 (1.5%) | 0.161 |
| Diabetes mellitus | 42 598 (3.7%) | 5478 (11.2%) | 0.289 | 2638 (19.5%) | 0.510 |

Continued

**Table 1** Continued

| | Non-exposed (n=1 151 347) | Any prior GC exposure* (n=48 806) | | High GC exposure† (n=13 497) | |
|---|---|---|---|---|---|
| | | | SMD versus non-exposed | | SMD versus non-exposed |
| Rheumatic disease | 8727 (0.8%) | 6568 (13.5%) | 0.510 | 4109 (30.4%) | 0.897 |
| Cancer | 44 717 (3.9%) | 7666 (15.7%) | 0.406 | 3511 (26.0%) | 0.653 |
| Prescribed medications | | | | | |
| Inhaled corticosteroids | 120 572 (10.5%) | 17 815 (36.5%) | 0.645 | 4258 (31.5%) | 0.536 |
| Oral anticoagulants | 57 018 (5.0%) | 8693 (17.8%) | 0.413 | 3992 (29.6%) | 0.689 |
| Statins | 101 469 (8.8%) | 11 096 (22.7%) | 0.389 | 4556 (33.8%) | 0.640 |
| ACEIs | 67 321 (5.8%) | 6969 (14.3%) | 0.283 | 2860 (21.2%) | 0.460 |
| Opioids | 112 707 (9.8%) | 12 505 (25.6%) | 0.424 | 4523 (33.5%) | 0.601 |

Data are n (%) or mean (SD).
*One or more prescriptions of oral GCs within 12 months before the COVID-19 infection date.
†Two or more prescriptions and prednisolone ≥750 mg or equivalent amount of betamethasone, dexamethasone, methylprednisolone, prednisone, hydrocortisone, cortisone or deflazacort within 6 months before COVID-19 infection date. Note this group is a subset of any prior exposure.
ACEI, ACE inhibitor; COPD, chronic obstructive pulmonary disease; CV, cardiovascular; GC, glucocorticoid; SMD, standardised mean difference.

association for high exposure with death from myocardial infarction was similar in strength to that with fatal stroke, but marginally not significant (figure 3).

### Association between GC exposure and severe COVID-19

Compared with the non-exposed group, the high exposure group demonstrated an elevated risk of hospitalisation for COVID-19 (adjusted HR 1.51, 95% CI 1.46 to 1.57) as well as ICU admission (1.40, 1.26 to 1.55) (figure 3).

There were also consistent but relatively weaker effects observed in patients with any prior exposure.

### Alternative analysis approach with propensity score matching

Propensity score matching resulted in 91 878 non-exposed individuals matched to 47 737 individuals with any exposure and 23 898 individuals matched to 12 943 individuals with high exposure (online supplemental table S4). The numbers of outcome events in the matched

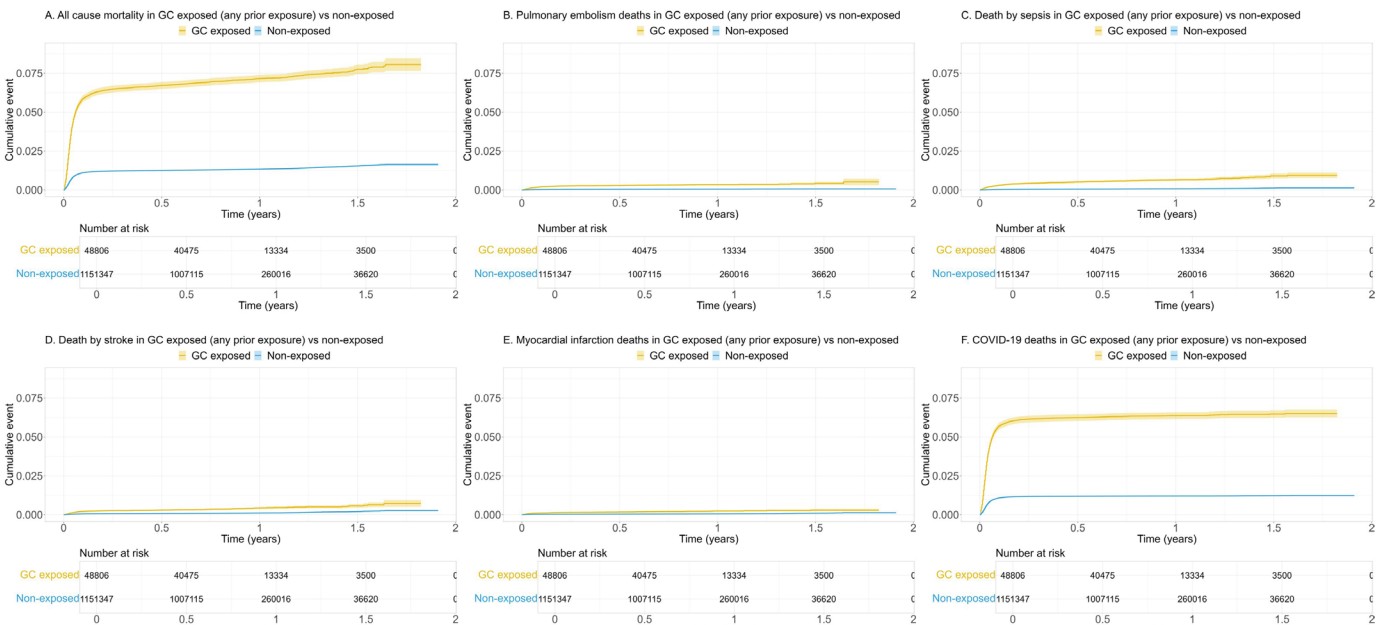

**Figure 1** Unadjusted cumulative incidence of all-cause (A) and cause-specific (B–F) mortality among COVID-19 patients in Sweden with any prior exposure* to oral glucocorticoids compared with non-exposed, from COVID-19 infection date to 30 November 2021. GC, glucocorticoids. *Any prior GC exposure equals ≥1 prescription of oral GC within 12 months before the COVID-19 infection date.

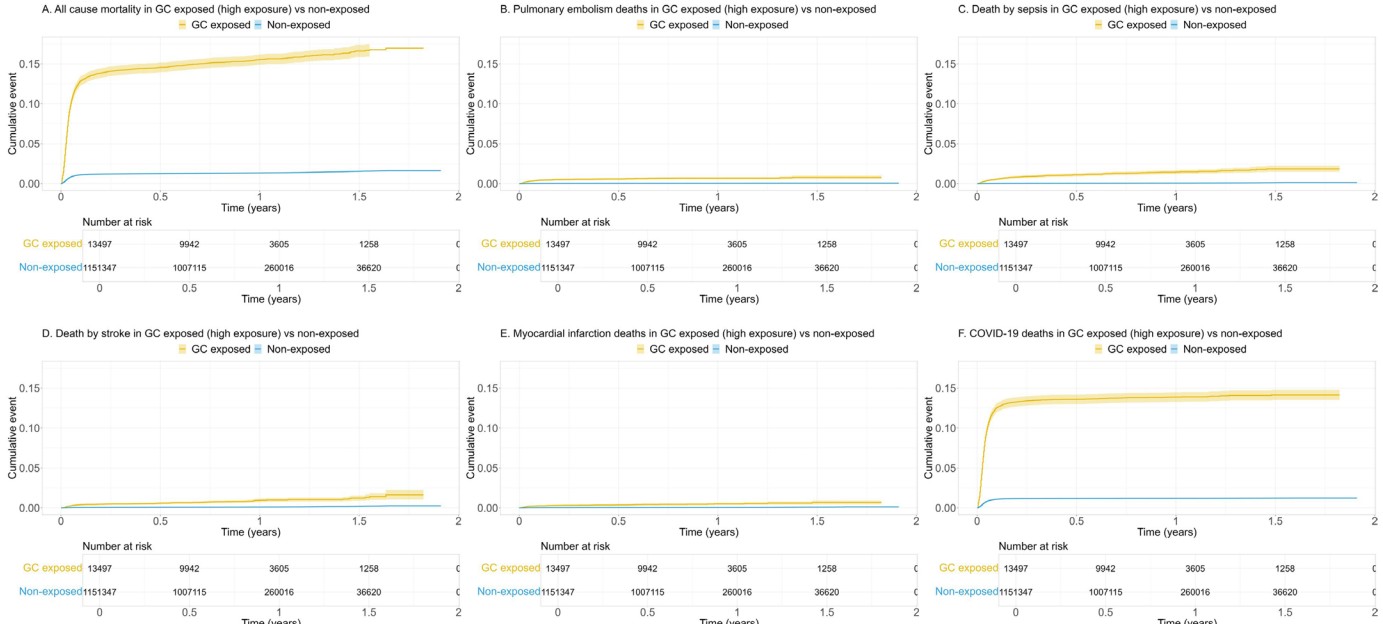

**Figure 2** Unadjusted cumulative incidence of all-cause (A) and cause-specific (B–F) mortality among COVID-19 patients with high exposure* to oral glucocorticoids compared with non-exposed, from COVID-19 infection date to 30 November 2021. GC, glucocorticoids. *High GC exposure equals ≥2 prescriptions with a total of ≥750 mg of prednisolone or equivalent within 6 months before COVID-19 infection date. This group is a subset of any prior exposure (see figure 1).

cohorts are indicated in online supplemental table S5. In the matched cohorts, the covariates of the exposed and matched non-exposed groups were quite well balanced (online supplemental table S6) with good overlap of the propensity scores (online supplemental figure S1) and low absolute SMDs between the matched cohorts (online supplemental tables S5 and S6, figure S2).

The results obtained from the propensity score-matched analysis were, overall, in good agreement with the adjusted Cox model (figure 3). Propensity score matching indicated somewhat stronger effects of GC exposure on deaths due to pulmonary embolism and myocardial infarction, and weaker effects on deaths due to sepsis and stroke, severe COVID-19 and overall mortality in both high exposure and any exposure categories (figure 3). In this analysis, high oral GC exposure was not significantly associated with death from stroke but, conversely, it was significantly and more strongly associated with deaths caused by myocardial infarction.

In a post hoc sensitivity analysis, we sought to address potential confounding due to slight remaining covariate imbalances through adjustment and propensity score matching, using an expanded set of covariates. This led to improved balance across most covariates (online supplemental table S7). Overall, the findings closely aligned with the main analysis except that ICU admission for COVID-19 was less common in GC users in this analysis (online supplemental table S8).

## DISCUSSION
This nationwide population-based cohort study of approximately 1.2 million individuals who had contracted COVID-19 infection showed increased mortality following the infection for 48 806 oral GC users compared with non-users, especially from sepsis, stroke and pulmonary embolism. Individuals receiving oral GC treatment prior to the infection also had an increased risk of COVID-19-related outcomes including hospitalisation, ICU admission and death. Furthermore, the results were robust in sensitivity analyses with extended confounding adjustment.

### Comparison with other studies
Our study showed an increased risk of death from sepsis after contracting COVID-19 and the risk was greater in the high GC exposure group. Treatment with GCs is known to cause immunosuppression and GC users have increased susceptibility to infections.[23] Similarly, studies on endogenous hypercortisolism and GC users show a high mortality risk due to infection in other settings.[2 24] For example, one-third of deaths in patients with Cushing's syndrome is due to infections[24] and GC users have generally increased mortality from sepsis and pneumonia.[2] There is also evidence that higher doses of GCs are associated with an increased risk of pneumonia and mortality in COVID-19 patients hospitalised with clinical hypoxia compared with standard low-dose GC treatment.[13] Our results are also in line with observational data on patients with rheumatic diseases, showing that prednisolone dose ≥10 mg/day prior to COVID-19 was associated with increased hospitalisation for COVID-19.[14]

We also found increased mortality from pulmonary embolism in GC users after contracting COVID-19. A generally increased incidence of thromboembolism in patients with endogenous hypercortisolism is known.[25] A recent study showed that patients with Cushing's

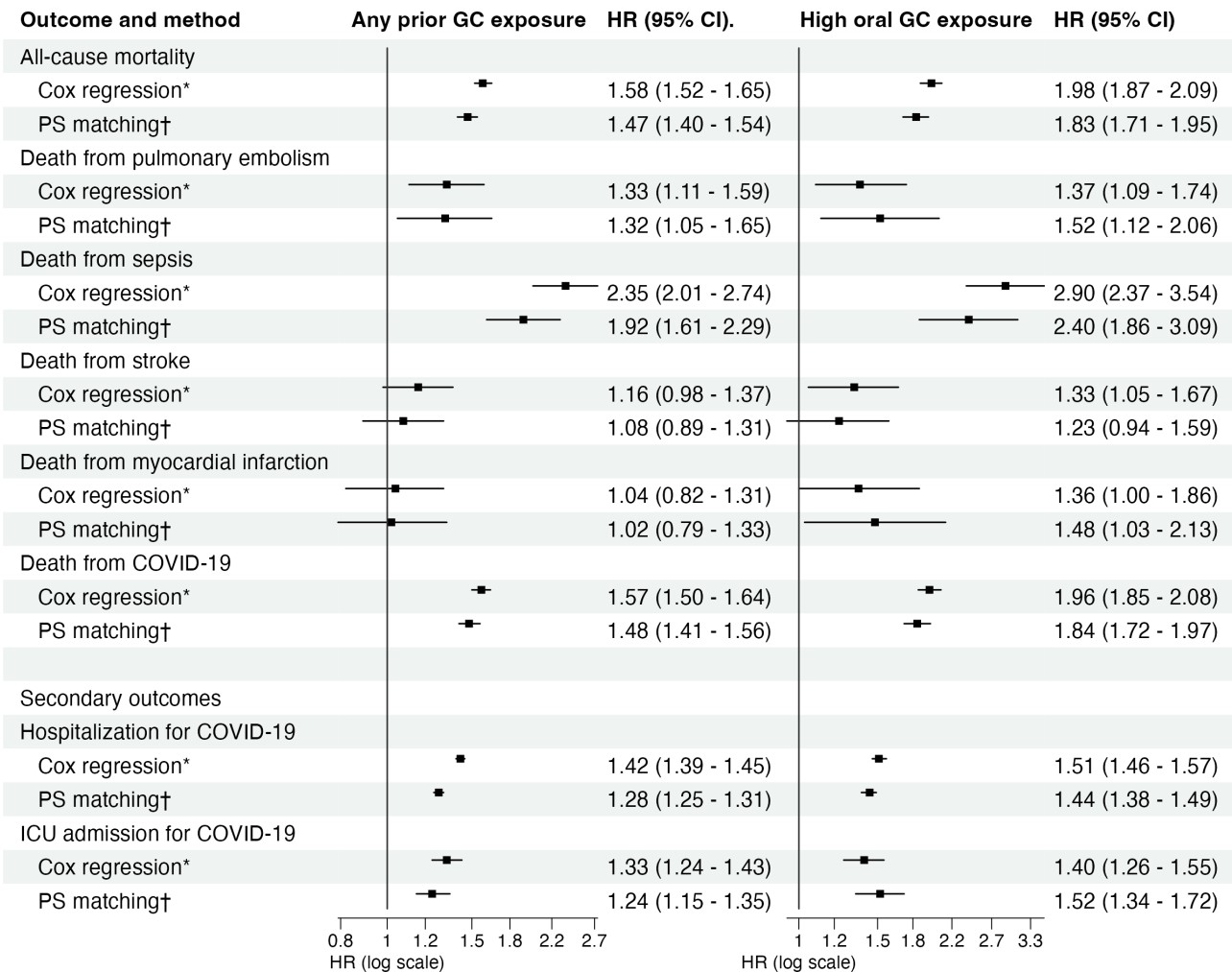

| Outcome and method | Any prior GC exposure | HR (95% CI). | High oral GC exposure | HR (95% CI) |
|---|---|---|---|---|
| **All-cause mortality** | | | | |
| Cox regression* | | 1.58 (1.52 - 1.65) | | 1.98 (1.87 - 2.09) |
| PS matching† | | 1.47 (1.40 - 1.54) | | 1.83 (1.71 - 1.95) |
| **Death from pulmonary embolism** | | | | |
| Cox regression* | | 1.33 (1.11 - 1.59) | | 1.37 (1.09 - 1.74) |
| PS matching† | | 1.32 (1.05 - 1.65) | | 1.52 (1.12 - 2.06) |
| **Death from sepsis** | | | | |
| Cox regression* | | 2.35 (2.01 - 2.74) | | 2.90 (2.37 - 3.54) |
| PS matching† | | 1.92 (1.61 - 2.29) | | 2.40 (1.86 - 3.09) |
| **Death from stroke** | | | | |
| Cox regression* | | 1.16 (0.98 - 1.37) | | 1.33 (1.05 - 1.67) |
| PS matching† | | 1.08 (0.89 - 1.31) | | 1.23 (0.94 - 1.59) |
| **Death from myocardial infarction** | | | | |
| Cox regression* | | 1.04 (0.82 - 1.31) | | 1.36 (1.00 - 1.86) |
| PS matching† | | 1.02 (0.79 - 1.33) | | 1.48 (1.03 - 2.13) |
| **Death from COVID-19** | | | | |
| Cox regression* | | 1.57 (1.50 - 1.64) | | 1.96 (1.85 - 2.08) |
| PS matching† | | 1.48 (1.41 - 1.56) | | 1.84 (1.72 - 1.97) |
| | | | | |
| **Secondary outcomes** | | | | |
| **Hospitalization for COVID-19** | | | | |
| Cox regression* | | 1.42 (1.39 - 1.45) | | 1.51 (1.46 - 1.57) |
| PS matching† | | 1.28 (1.25 - 1.31) | | 1.44 (1.38 - 1.49) |
| **ICU admission for COVID-19** | | | | |
| Cox regression* | | 1.33 (1.24 - 1.43) | | 1.40 (1.26 - 1.55) |
| PS matching† | | 1.24 (1.15 - 1.35) | | 1.52 (1.34 - 1.72) |

0.8 1 1.2 1.5 1.8 2.2 2.7
HR (log scale)

1 1.2 1.5 1.8 2.2 2.7 3.3
HR (log scale)

COVID-19 = Coronavirus disease 2019; oral GC = oral glucocorticoids; adj.HR = adjusted hazard ratio; PS = propensity score;ICU = intensive care unit.
Covariates = age, sex, education, employment, and previous medical history (diabetes, deep vein thrombosis, pulmonary embolism, hypertension, stroke, ischemic heart disease, heart failure, cancer, chronic obstructive pulmonary disease , rheumatologic conditions, oral anticoagulation therapy; * = adjusted for covariates; †PS matched on covariates;
Note: Unadjusted estimates are mentioned in supplementary table S8.

**Figure 3** Adjusted Cox regression analysis and PS-matched analysis comparing the risk of all-cause and cause-specific mortality between COVID-19 patients with prior exposure to oral glucocorticoids and non-exposed patients. GC, glucocorticoids; ICU, intensive care unit; PS, propensity score. Covariates=age, sex, education, employment and previous medical history (diabetes, deep vein thrombosis, pulmonary embolism, hypertension, stroke, ischaemic heart disease, heart failure, cancer, chronic obstructive pulmonary disease, rheumatological conditions and oral anticoagulation therapy. Any prior GC exposure equals ≥1 prescription of oral GC within 12 months before the COVID-19 infection date. High GC exposure equals ≥2 prescriptions with a total of ≥750 mg of prednisolone or equivalent within 6 months before COVID-19 infection date. The latter group is a subset of any prior exposure. *Adjusted for covariates; †PS matched on covariates. Note: Unadjusted estimates are mentioned in online supplemental table S8.

syndrome have a generally increased incidence of stroke and thromboembolism.[25] Studies on GC users have reported similar findings with an approximately twofold increased risk of both pulmonary embolism and deep vein thrombosis.[26] The risk of comparable events after COVID-19 in the current study is similar in magnitude.

GC treatment was not significantly associated with death from stroke in our study, but it was significantly and more strongly associated with deaths due to myocardial infarction. The association was stronger in the high exposure group compared with the any prior exposure group. Our findings are in line with a population-based,

nested, case-control study from the UK on oral GC and risk of cardiovascular and cerebrovascular disease based on data from 1988 to 1998.[27] The study showed increased risk of ischaemic heart disease (OR 1.20, 95% CI 1.11 to 1.29) in current GC users, but not for increased risk of stroke/transient ischaemic attack (OR 0.95, 95% CI 0.89 to 1.01).[27]

We used both adjusted Cox models and propensity score matching to control for potential confounders in an effort to obtain unbiased estimates of the impact of oral GC use on severe outcomes. To clarify and understand imbalances in characteristics between the exposure groups

before and after propensity score matching, we evaluated SMDs and achieved good balance after matching.[28] It may be noted that the estimand for the adjusted Cox model is the conditional average treatment effect for the treated (ATE), that is, the effect if everyone were to be on treatment with oral GCs, conditional on their characteristics, while the estimand for the matched analysis is the average treatment effect for the treated (ATT), that is, the effect among those actually receiving oral GC treatment.[29] ATE can be used to decide whether to mandate treatment for all patients, whereas ATT can be used to decide whether to withhold it from patients who would otherwise receive it. Nevertheless, the two methods produced comparable results that are clinically relevant and complementary.

### Strengths and limitations of this study

This study has several strengths including a population-level sample, a large number of covariates and a large comparison cohort which minimises the risk of selection bias. However, since our study is observational, the analyses may still be subject to residual bias and unmeasured confounding, and thus the results require cautious interpretation. All GC users have an underlying disease that may increase mortality risk, potentially confounding the results. We mitigated this concern by adjusting for carefully preselected covariates, by conducting a sensitivity analysis with expanded covariate selection, and by using two analytical approaches. The results of all analyses were fairly consistent, supporting the robustness of the findings. The sensitivity analyses, however, suggested that oral GC per se may not be the driver for increased COVID-19-related ICU admission, although this is speculative and should be interpreted with caution due to the more limited power for this rare outcome and the risk for overadjustment in these analyses.

With regard to stage of severity and exposure to GC, we assessed both 'any' exposure and 'high' exposure to address differences in the severity of the underlying condition warranting oral GC treatment. In our study, we relied on a prescription register that only provided information about dispensed prescriptions, and thus adherence to treatment and actual use must be assumed (as is generally the case). Our study exclusively focused on the oral administration of GCs and their impact on COVID-19. While inhaled GCs have been studied in Swedish context,[30] our study places a deliberate emphasis on the oral route. Other administration routes for GC, such as topical and parenteral administrations, were intentionally excluded from our investigations, as they are likely less directly pertinent to COVID-19.

### Unanswered questions and future research

The causal relationship between GC use and mortality remains a topic yet to be fully elucidated, primarily due to the fact that all long-term GC users have an underlying disease that may heighten mortality risk. Our study provides a comprehensive overview of excess mortality in this population. However, it is crucial to highlight that studying each underlying medical condition necessitates specific analytical considerations. To enhance our understanding of the factors contributing to excess mortality, future research should prioritise dissecting the precise impact of prolonged GC use on these underlying conditions. Our mortality estimates essentially represent a weighted average effect over the pandemic period, primarily for power and precision reasons. Larger cohorts in future studies may help reveal wave-specific effects.

Fundamentally, GC treatment suppresses the hypothalamic-pituitary-adrenal axis and can cause GC-induced adrenal insufficiency.[4 31] Undiagnosed and untreated GC-induced adrenal insufficiency can lead to a life-threatening adrenal crisis. To prevent adrenal crisis, GC users should know how to increase their GC dose during sick days and during intercurrent illness.[32] Awareness of GC-induced adrenal insufficiency is increasing. In 2020, guidance on the prevention of adrenal crisis during the COVID-19 pandemic was published and recommended that GC users increase their daily GC dose during COVID-19 infection and to take GC every 6–12 hours (depending on the type of GC used).[32] GC users who need to be admitted to hospital because of COVID-19 should receive GC intravenously.[32]

### CONCLUSION

Patients receiving GC treatment prior to COVID-19 infection face excess mortality in the aftermath of their acute COVID-19 infection, particularly from sepsis, pulmonary embolism and COVID-19. These patients also have a high risk of ICU admission and death due to COVID-19, and close monitoring and clinical attention may thus prevent their death.

**Author affiliations**
[1]Department of Internal Medicine and Clinical Nutrition, University of Gothenburg, Goteborg, Sweden
[2]Department of Endocrinology, Sahlgrenska University Hospital, Goteborg, Sweden
[3]School of Public Health and Community Medicine, Institute of Medicine, Sahlgrenska Academy, University of Gothenburg, Goteborg, Sweden
[4]Cardiovascular, Renal and Metabolism (CVRM), BioPharmaceuticals R&D, AstraZeneca, Gothenburg, Sweden
[5]Wallenberg Center for Molecular and Translational Medicine, University of Gothenburg, Gothenburg, Sweden

**Acknowledgements** We thank Peter Todd of Tajut Ltd. (Kaiapoi, New Zealand) for third-party editorial support, for which he received financial compensation from ALF funding.

**Contributors** OR, MJE, BKK, FN, HL, DO and GJ were involved in the conceptualisation and design of the study. FN and HL acquired the data. BK and HL performed statistical analysis. OR, MJE, BKK, FN, HL, DO and GJ were involved with interpretation of the data and initial drafting of the manuscript. OR and FN are the guarantors. All authors participated in the critical appraisal, further interpretation, revision and approved the final manuscript.

**Funding** ALF agreement (ALFGBG-938453, ALFGBG-971130, ALFGBG-978954, ALFGBG-966066), Swedish Research Council (2020-02828, 2019-01112) and Swedish Heart-Lung Foundation (20210030, 20210581).

**Competing interests** GJ has served as consultant for Ascendis Pharma, Astra Zeneca and Novo Nordisk, and has received lecture fees from Novo Nordisk and Pfizer. DO has received unrestricted project grants from Pfizer and is an employee

at AstraZeneca as of 8 August 2021. FN holds some AstraZeneca shares. BKK, ME, HL and OR have nothing to disclose.

**Patient and public involvement** Patients and/or the public were not involved in the design, or conduct, or reporting, or dissemination plans of this research.

**Patient consent for publication** Not applicable.

**Ethics approval** Ethical approval for this study was obtained from the Swedish Ethical Review Authority (2020-01800 with subsequent amendments).

**Provenance and peer review** Not commissioned; externally peer reviewed.

**Data availability statement** Data may be obtained from a third party and are not publicly available. The individual-level data used in this study are pseudonymised and sourced from Swedish healthcare registers. Interested researchers can obtain access to the data from the appropriate Swedish public data holders subject to obtaining ethics approval for the research and adhering to all relevant legislation, processes and data protection protocols.

**ORCID iDs**
Margret J Einarsdottir http://orcid.org/0000-0002-6010-9599
Brian Kibiwott Kirui http://orcid.org/0000-0002-6656-6029
Huiqi Li http://orcid.org/0000-0002-1127-0829
Daniel Olsson http://orcid.org/0000-0002-9734-0786
Gudmundur Johannsson http://orcid.org/0000-0003-3484-8440
Fredrik Nyberg http://orcid.org/0000-0003-0892-5668
Oskar Ragnarsson http://orcid.org/0000-0003-0204-9492

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
