## [Reviewer comments · BMJ Open]

ARTICLE DETAILS

TITLE (PROVISIONAL)	Impact of chronic oral glucocorticoid treatment on mortality in patients with COVID-19: analysis of a population-based cohort
AUTHORS	Ragnarsson, Oskar; Einarsdottir, Margret; Kibiwott Kirui, Brian; Li, Huiqi; Olsson, Daniel; Johannsson, Gudmundur; Nyberg, Fredrik

VERSION 1 – REVIEW

REVIEWER	Dregan, Alexandru King's College London, UK, Primary Care and Public Health Sciences
REVIEW RETURNED	23-Oct-2023

GENERAL COMMENTS	The work describes the association between oral GC with mortality in a large population with a Covid-19 infection in Sweden. The ms is well-written and the methodology is appropriate for the study aims. This Reviewer has some minor clarifications for the authors. 1. What about those participants that were prescribed GC's prior to 12 months at-risk period?2. Do authors have access to regional data or ethnicity data to provide general evidence of potential inequalities in mortality?3. Were there differences in mortality across the different waves of the pandemic?4. For non-Covid related mortality the best comparison group would have been those without a Covid-19 infection.
---

REVIEWER	Beladiya, Jayesh V. LM College of Pharmacy
REVIEW RETURNED	28-Oct-2023

GENERAL COMMENTS	It is good study but kindly address the below query; 1. How was the route of administration, dose and pulse administration considered?2. How the stage of severity category and exposure to GC at stage of severity considered in the study?3. How the assess the presence of other medical condition and relation to GC?4. How the assess the effect of concomitant medication?
--

VERSION 1 – AUTHOR RESPONSE

Reviewer: 1

Dr. Alexandru Dregan, King's College London, UK **Comments to the Author:**

The work describes the association between oral GC with mortality in a large population with a Covid-19 infection in Sweden. The ms is well-written and the methodology is appropriate for the study aims. This Reviewer has some minor clarifications for the authors.

Thank you for the overall positive assessment.

1. What about those participants that were prescribed GC's prior to 12 months atrisk period?

In this study, our goal was to capture recent or ongoing orally administered GC and identify individuals with exposure before entering the risk window. We used two register-based definitions of exposure: receiving at least one prescription in the year before the index (indicating recent use of oral GC), as well as requiring 2 prescriptions and a moderate cumulative dose in the prior year (indicating more consistent medication use during that time, and more likely extending into the follow-up period). As such, these definitions seek to identifying patients at risk when follow-up starts. Prescriptions before this 12-month period were not considered of relevance as exposure for the risk follow-up. We have now added some additional text in the Methods section:

lines 209 : "Recent or ongoing oral GC treatment was defined in two ways"

and 218-234. "These exposure definitions target recent use of oral GC, and for the "high exposure" more consistent and recent medication use, more likely extending into the future follow-up period, thus sharpening the identification of regular users and as such, patients at risk, and ensuring robust data analysis."

2. Do authors have access to regional data or ethnicity data to provide general evidence of potential inequalities in mortality?

This is an interesting aspect of the COVID-19 pandemic. Unfortunately, we do not have access to regional or ethnicity data. In addition, ethnic inequalities in mortality were not the focus of our analysis and are best addressed elsewhere. The available information of this type in our study is limited to birth country, and as such, we are unable to provide specific evidence regarding potential inequalities in mortality based on regional or ethnic factors. Birth country was not considered to be a likely confounder or effect modifier of the association between oral GC and mortality, and so was not included in this study, based on our preselection of likely relevant covariates.

3. Were there differences in mortality across the different waves of the pandemic?

This is an interesting comment, and one that the authors considered when designing the study.

Given our focus on assessing the impact of oral glucocorticoids on overall and causespecific mortalities of interest, we chose not to conduct a wave-specific analysis. Although oral glucocorticoid use is relatively common in the population, subdividing mortality within the group across different waves would reduce the statistical power and increase imprecision of the results. Therefore, we present mortality estimates from the Cox regression over the entire pandemic period, essentially providing a weighted average of the effect over all waves. This approach supports clinical utility and

interpretability while aligning with the primary objectives of our study. We appreciate the comment, and have incorporated a corresponding clarification in the “unanswered questions and future research” section to address this remark.

Lines 438-441: “Our mortality estimates essentially represent a weighted average effect over the pandemic period, primarily for power and precision reasons. Larger cohorts in future studies may help reveal wave-specific effects.”

4. For non-Covid related mortality the best comparison group would have been those without a Covid-19 infection.

Thank you for this comment. While we acknowledge the potential importance of a nonCOVID comparison group, our study specifically focused on evaluating the impact of oral glucocorticoids (GC) in COVID-19 patients, i.e. conditional of being infected. Our aim was to assess the specific relationship between oral GC use and mortality within this cohort. A comparison with a non-Covid group would address a different question, and mortality differences would be potentially due to both oral GC exposure and Covid-19 infection. We have previously done a study in the general population to investigate mortality in oral GC users compared to controls who did not use oral GC (Einarsdottir et al, Front Endocrinol. 2022) and the results showed increased mortality in oral GC users compared to the background population (HR 2.1, 95% CI 2.0-2.1). In the current study, the hypothesis was that patients on oral GC treatment before being diagnosed with COVID19 have an increased overall and specific mortality from various causes (pulmonary embolism, myocardial infarction, stroke, sepsis and COVID-19). To test this hypothesis, we investigated the association between prior GC treatment and the risk of mortality from these causes among COVID-19 patients. We have slightly modified the wording of the study aim in the abstract:

(line 79: “We investigated if patients with oral GC use prior to COVID-19 had increased overall mortality overall and by selected causes.”)

to align with this, and also emphasize this point in a comment on potential overadjustment in Methods:

(line 274-279: Sensitivity analyses were also conducted incorporating additional potential confounders of interest, in particular greater detail regarding the underlying conditions that are indications for oral GC. This sensitivity analysis could, however, instead suffer from potential overadjustment in relation to the study hypothesis of understanding the overall increased risk patients in these groups on oral GC faced when infected with COVID-19”)

when describing the added sensitivity analysis (see also responses below)

Reference

Einarsdottir MJ, Ekman P, Molin M, Trimpou P, Olsson DS, Johannsson G, Ragnarsson O. High mortality rate in oral glucocorticoid users: a population-based matched cohort study. Front Endocrinol (Lausanne). 2022;13:918356.

Reviewer: 2

Dr. Jayesh V. Beladiya, LM College of Pharmacy Comments to the Author:

It is good study but kindly address the below query;

1. How was the route of administration, dose and pulse administration considered?

Thank you for the opportunity to elaborate on this. Our study exclusively focuses on the oral administration of glucocorticoids (GCs) in addressing a distinct research gap associated with oral GCs and their impact on COVID-19. While inhaled GCs have been studied in the Swedish context (Labor et al., 2023), our study places a deliberate emphasis on the oral route. We recognize the availability of other administration methods for GCs, such as topical creams and parenteral injections, but these

alternative routes were intentionally excluded from our investigation, as they may be less directly pertinent to COVID-19. Our study maintains a focused approach, aiming to comprehend the specific effects of oral GC usage in COVID-19 patients, and we have now provided additional comments on this in the Discussion section:

lines 424-428: *“Our study exclusively focused on the oral administration of GCs and their impact on COVID-19. While inhaled glucocorticoids have been studied in Swedish context (28), our study places a deliberate emphasis on the oral route. Other administration routes for GC, such as topical and parenteral administrations, were intentionally excluded from our investigations, as they are likely less directly pertinent to COVID-19.”*

Reference:

Labor, M., Kirui, B. K., Nyberg, F., & Vanfleteren, L. E. G. W. (2023). Regular Inhaled Corticosteroids Use May Protect Against Severe COVID-19 Outcome in COPD. *International Journal of Chronic Obstructive Pulmonary Disease*, 18, 1701–1712. <https://doi.org/10.2147/COPD.S404913>

2. How the stage of severity category and exposure to GC at stage of severity considered in the study?

Thank you for this comment. In addressing the stage of severity and exposure to GC, we aimed to differentiate between "any" exposure and "high" exposure to precisely address differences in the severity of the underlying condition warranting oral GC treatment. The distinction between "any" and "high" exposure thus allowed us to explore nuanced variations in the impact of oral GC across different outcomes. Regarding severity of COVID-19, we looked at severe COVID-19 by focusing on hospitalization for COVID-19 and those who died of COVID-19. Further clarification on this point has been added to: Methods lines 218-222 *“These exposure definitions target recent use of oral GC, and for the “high exposure” more consistent and recent medication use, more likely extending into the future follow-up period, thus sharpening the identification of regular users and as such, patients at risk, and ensuring robust data analysis.”*, and

Discussion, lines 419-421 *“With regard to stage of severity and exposure to GC, we assessed both “any” exposure and “high” exposure to address differences in the severity of the underlying condition warranting oral GC treatment. In our study, we relied on a prescription register that only provided information about dispensed prescriptions, and thus adherence to treatment and actual use must be assumed (as is generally the case).”*

3. How the assess the presence of other medical condition and relation to GC?

This is an important comment. We assessed and adjusted for multiple comorbidities (diabetes, deep vein thrombosis, pulmonary embolism, hypertension, stroke, ischaemic heart disease, heart failure, oral anticoagulation therapy, cancer, chronic obstructive pulmonary disease, rheumatologic conditions), based on a *priori* selection.

Since GC users have a wide range of underlying diseases that may increase mortality risk, potentially confounding the analyses, we conducted a sensitivity analysis with further adjustment for additional potential confounding factors, in particular greater underlying conditions that are indications for oral GC, including adrenal insufficiency, rheumatoid arthritis, chronic lower respiratory diseases, other respiratory diseases principally affecting the interstitium, non-infective enteritis and colitis, dermatitis and eczema, inflammatory polyarthropathies, systemic connective tissue disorders (excluding rheumatoid arthritis), glomerular diseases, organ or tissue transplant, chronic liver disease, and some

further covariates including medication usage such as inhaled glucocorticoids, statins, ACE inhibitors, opioids, as well as factors like alcoholism and substance abuse.

This analysis provided further improved balance across most covariates. Overall, the findings closely aligned with the main analysis. except that ICU admission for COVID-19 was less common in GC users in this analysis. This sensitivity analysis could, however, instead suffer from potential overadjustment in relation to the study hypothesis of understanding the overall increased risk patients in these groups on oral GC faced when infected with Covid-19.

We have added additional text to explain this sensitivity analysis in Methods: (lines 249-257: "*In sensitivity analyses, we considered additional covariates : adrenal insufficiency, rheumatoid arthritis, chronic lower respiratory diseases, other respiratory diseases principally affecting the interstitium, non-infective enteritis and colitis, dermatitis and eczema, inflammatory polyarthropathies, systemic connective tissue disorders (except rheumatoid arthritis), glomerular diseases, organ or tissue transplant , chronic liver disease, medications: inhaled corticosteroids, statins, angiotensin converting enzyme inhibitors (ACEIs), opioids, alcoholism and other substance abuse*"),

lines 274-279: "*Sensitivity analyses were also conducted incorporating additional potential confounders of interest, in particular greater detail regarding the underlying conditions that are indications for oral GC. This sensitivity analysis could, however, instead suffer from potential overadjustment in relation to the study hypothesis of understanding the overall increased risk patients in these groups on oral GC faced when infected with COVID-19.*"),

Results (lines 338-343 "*In a post-hoc sensitivity analysis, we sought to address potential confounding due to slight remaining covariate imbalances through adjustment and PS matching utilizing an expanded set of covariates. This led to improved balance across most covariates (Table S7). Overall, the findings closely aligned with the main analysis except that ICU admission for COVID-19 was less common in GC users in this analysis*") and

Discussion (lines 410-417 "*We mitigated this concern by adjusting for carefully preselected covariates, by conducting a sensitivity analysis with expanded covariate selection, and by using two analytical approaches. The results of all analyses were fairly consistent, supporting the robustness of the findings. The sensitivity analyses, however, suggests that oral GC per se may not be the driver for increased COVID19-related ICU admission, although this is speculative and should be interpreted with caution due to the more limited power for this rare outcome and the risk for overadjustment in these analyses.*")

4. How the assess the effect of concomitant medication?

Thank you for this comment. To evaluate the effect of concomitant medications, we considered information on various medications, including those with at least one filled prescription in the year prior to the index date. Specifically, the medications examined were inhaled glucocorticoids, oral anticoagulants, statins, ACEIs, and opioids, as detailed in Table 1.

In our main analysis, we chose to adjust for the use of oral anticoagulants as the most relevant to confound the association between glucocorticoids use and our mortality outcomes. In place of the other concomitant medications, we chose to focus instead on adjusting for comorbidities (diabetes, deep vein thrombosis, pulmonary embolism, hypertension, stroke, ischaemic heart disease, heart failure, cancer).

In our sensitivity analysis mentioned in the previous response, where we included a broader set of covariates and in particular greater detail regarding the underlying conditions that are indications for oral GC, we also included some additional potentially relevant medications, aiming for further

improved balance across all covariates. As noted, the results were highly consistent with the main analysis, supporting the robustness of our findings.

VERSION 2 – REVIEW

REVIEWER	Dregan, Alexandru King's College London, UK, Primary Care and Public Health Sciences
REVIEW RETURNED	16-Feb-2024

GENERAL COMMENTS	I am satisfied with the reviewers responses to my initial concerns and I have no further queries. Congratulations on a very interesting analysis.
---

REVIEWER	Beladiya, Jayesh V. LM College of Pharmacy
REVIEW RETURNED	07-Feb-2024

GENERAL COMMENTS	It is very interesting and useful study. However, I have queries as mentioned below;  1. On which basis or guideline the author decide the higher and less exposure to GC. 2. What is the role of the usage of the prophylaxis antibiotics with GC to avoid secondary infection. It should be discussed in the limitations or scope. 3. Resolution of the image can be improved if possible.
--

VERSION 2 – AUTHOR RESPONSE

Reviewer: 2

Dr. Jayesh V. Beladiya, LM College of Pharmacy

Comments to the Author:

It is very interesting and useful study. However, I have queries as mentioned below;

1. On which basis or guideline the author decide the higher and less exposure to GC.

The basis for high exposure to glucocorticoids (GC) in the study was arbitrary, and primarily informed by the fact that doses exceeding 5mg of prednisolone are considered supraphysiological and have been associated with worse outcomes compared to doses of 5mg and below (Movahedi, M., et al. (2016)). On this basis, low-dose glucocorticoid treatment would be defined as a dose lower than 5 mg prednisolone per day. Consequently, in our study, high exposure was defined as prednisolone ≥ 750 mg within 6 months before the index date, i.e. approximately $>4-5$ mg of prednisolone per day which is a supraphysiological GC dose. By computing the average prednisolone dosage per day over this period (750 mg/180 days), high exposure equates to a dose of $\geq 4-5$ mg prednisolone per day.

We have now added clarified in the methods and added the reference lines 223-226.

“High exposure was defined as prednisolone ≥ 750 mg within 6 months before the index date, i.e. approximately ≥ 4 -5 mg of prednisolone per day which is supraphysiological dose of GC (20). Doses below 5 mg of prednisolone per day have not been associated with an increased risk of death (21).

Reference:

Movahedi, M., Costello, R., Lunt, M. *et al.* Oral glucocorticoid therapy and all-cause and cause-specific mortality in patients with rheumatoid arthritis: a retrospective cohort study. *Eur J Epidemiol* **31**, 1045–1055 (2016). <https://doi.org/10.1007/s10654-016-0167-1>

2. What is the role of the usage of the prophylaxis antibiotics with GC to avoid secondary infection. It should be discussed in the limitations or scope.

This is an important consideration. GC users have increased risk of infections (Stuck AE) and therefore prophylactic antibiotics could potentially reduce the risk of secondary infections. However, this strategy is disease-specific and unrelated to GC usage. For instance, in patients with chronic obstructive pulmonary disease (COPD) patients taking oral GC, prophylactic antibiotic therapy is reserved for those with presumed bacterial infections, weighing factors like antibiotic resistance and adverse effects (Viniol et al). In a COVID-19 study, antibiotic prophylaxis did not reduce secondary bacterial infections but increased multidrug-resistant bacteria colonization (Membrillo de Novales FJ, et al). Despite potential benefits, prophylactic antibiotics are uncommon among GC users in Sweden, hence excluded from our analysis.

No changes regarding this comment were made in the manuscript.

References:

Stuck AE, Minder CE, Frey FJ. Risk of Infectious Complications in Patients Taking Glucocorticosteroids. *Rev Infect Dis* (1989) 11(6):954–63.

Membrillo de Novales FJ, Ramírez-Olivencia G, Mata Forte MT, Zamora Cintas MI, Simón Sacristán MM, Sánchez de Castro M, Estébanez Muñoz M. The Impact of Antibiotic Prophylaxis on a Retrospective Cohort of Hospitalized Patients with COVID-19 Treated with a Combination of Steroids and Tocilizumab. *Antibiotics* (Basel). 2023 Oct 6;12(10):1515. doi: 10.3390/antibiotics12101515. PMID: 37887216; PMCID: PMC10604609.

Viniol C, Vogelmeier CF. Exacerbations of COPD. *Eur Respir Rev*. 2018 Mar 14;27(147):170103. doi: 10.1183/16000617.0103-2017. PMID: 29540496; PMCID: PMC9488662.

3. Resolution of the image can be improved if possible.

Figures 1 and 2 have now been improved with better resolution and improved legibility of the text. Figure 3 and the supplementary figures have relatively good resolution.

Reviewer: 1

Dr. Alexandru Dregan, King's College London, UK

Comments to the Author:

I am satisfied with the reviewers responses to my initial concerns and I have no further queries. Congratulations on a very interesting analysis.

Thank you for your positive assessment